# High-Grade Ferronickel Concentrates Prepared from Laterite Nickel Ore by a Carbothermal Reduction and Magnetic Separation Method

**DOI:** 10.3390/ma16227132

**Published:** 2023-11-11

**Authors:** Jingzhe Zhang, Chang Cao, Zhengliang Xue, Faliang Li, Shaoping Li, Hongjuan Duan, Haijun Zhang

**Affiliations:** 1The State Key Laboratory of Refractories and Metallurgy, Wuhan University of Science and Technology, Wuhan 430081, China; zhangjingzhe970518@163.com (J.Z.); 15337178025@163.com (C.C.); xuezhengliang@wust.edu.cn (Z.X.); lfliang@wust.edu.cn (F.L.); 2Hubei Three Gorges Laboratory, Yichang 443008, China; lisp@sinophorus.com

**Keywords:** high-grade ferronickel concentrate, laterite nickel ore, carbothermal reduction, magnetic separation

## Abstract

Nickel is widely used in industrial processes and plays a crucial role in many applications. However, most of the nickel resource mainly exists as nickel oxide in laterite nickel ore with complex composition, resulting in difficulty in upgrading the nickel content using physical separation methods. In this study, high-grade ferronickel concentrates were obtained through a carbothermal reduction and magnetic separation using laterite nickel ore and anthracite as raw materials. The effects of different types of additives (CaF_2_, Na_2_SO_4_, and H_3_BO_3_), carbon ratio (the molar ratio of oxygen atoms in the laterite nickel ore to carbon atoms in anthracite), and grinding time on the recoveries and grades of ferronickel concentrates were experimentally investigated, along with the microstructure and chemical composition of the products. CaF_2_ was proved to be the primary active additive in the aggregation and growth of the ferronickel particles and the improvement of the grade of the product. Under the optimal conditions of CaF_2_ addition of 9.85 wt%, carbon ratio of 1.4, and grinding time of 240 s, high-grade magnetically separable ferronickel concentrate with nickel grade 8.93 wt% and iron grade 63.96 wt% was successfully prepared. This work presents a practical method for the highly efficient recovery and utilization of iron and nickel from low-grade laterite nickel ore, contributing to the development of strategies for the sustainable extraction and utilization of nickel resources.

## 1. Introduction

As one of the important strategic metals, nickel is widely used in the stainless steel and new material industries owing to its superior mechanical strength, high chemical stability, good ductility, good catalytic activity, and strong corrosion resistance [1,2,3]. The nickel used in the stainless steel industry accounts for approximately 66.2% of the total consumption of nickel worldwide, resulting in dramatically increased demand for nickel in recent years [4]. In nickel production, most metal nickel is produced from nickel sulfide ores, and laterite nickel ore only accounts for about 40% due to its complex mineralogy, low nickel and iron grades, and the high treatment cost [5]. However, with the diminution of high-grade nickel sulfide ores, recovering nickel from laterite nickel ore as an alternative of nickel sulfide ores is drawing increasing attention since laterite nickel ore constitutes 72.2% of the world’s land-based nickel reserves [6,7,8]. Therefore, an economical method is urgently needed for recovering nickel from laterite nickel ore.

Laterite nickel ore is an important nickel mineral resource with large reserves, low grade, and difficult extraction. Nickel laterite deposits are mainly divided into limonite, transition layer, and garnierite according to their distribution in the nickel oxide deposit [9]. Among them, the upper layer of the deposit is limonitic nickel laterite ore with high iron content and low nickel content, which is often treated with a hydrometallurgical process; however, extracting nickel from limonitic nickel laterite ore is complex and requires sophisticated equipment [10]. The middle layer is the transition layer ore with relatively high iron, silicon, nickel, and magnesium content, therefore, the pyrometallurgical process or the hydrometallurgical process is suitable for transition layer ore treatment [11]. For the bottom saprolitic nickel laterite ore, it is difficult to concentrate nickel and iron through physical beneficiation due to its low nickel and iron content and the highly distributed garnierite [12]. With the export restriction and depletion of high-grade nickel laterite resources, the exploration and utilization of low-grade nickel laterite ore treatment possess broad development prospects owing to its rich reserves, low exploration cost, and mature smelting technology [13,14,15].

Nickel in laterite nickel ore exists mainly as silicate, indicating that a simple physical method is unable to extract nickel from the ore [16]. The current process approach for enriching ferronickel from laterite nickel ore is a pyrometallurgical process, such as the rotary kiln electric furnace process and blast furnace process [17]. The pyrometallurgical process has incomparable advantages in the quality of the ferronickel concentrate [18], nevertheless, it also suffers from high energy consumption, high demand for raw materials, and high volume of the waste stream [19,20]. To achieve carbon peak and neutrality targets and make full use of laterite nickel ore, the research of new technology with low carbon emissions and reduced energy consumption becomes the top priority [21,22]. Considering on the low grade of nickel and iron in laterite nickel ore, solid-state reduction roasting combined with magnetic separation to remove the magnesium silicates and unreduced oxides can be a promising method to produce a ferronickel concentrate from laterite nickel ore [23].

Herein, in this study, high-grade ferronickel concentrates were prepared by a carbothermal reduction and magnetic separation method from low-grade laterite nickel ore. The main influencing factors that could significantly influence the formation of ferronickel concentrate were investigated, including carbon ratio, types and amounts of additives, and grinding time. The effects of the simultaneous addition of multiple additives on the formation of melting phases, as well as their influences on the reduction of ferronickel alloys, were investigated in detail. This work provides guidance for effectively exploiting and utilizing low-grade laterite nickel ore to produce high-grade ferronickel concentrate with low energy consumption, which is of great significance in maintaining the nickel supply and holds promising potential in the application of blast furnace iron smelting, stainless steel production, alloy manufacturing, and electroplating industries.

## 2. Materials and Methods

### 2.1. Characterization and Thermodynamic Equilibrium Calculation Methods

The crushing process of the raw materials was conducted on a jaw crusher (XPC 125 × 100), and a 25 kW high-temperature carbon tube furnace (YHA-4035A, Shanghai Jule Industrial Development Co., Ltd. (Shanghai, China)) was employed for the carbothermal reduction process. The crystalline phases of the raw laterite nickel ore were identified via powder X-ray diffraction (XRD, X’Pert Pro, Philips, Amsterdam, The Netherlands) with Cu-Kα radiation (λ = 1.5406 Å) operated at 40 kV and 40 mA with a scanning rate of 10° (2θ) min^−1^, and the microstructures and elemental mappings of the samples were characterized using a field emission scanning electron microscope (FESEM, Novo 400, FEI Co., (Hillsboro, OR, USA)) equipped with an energy dispersive spectrometer (EDS, Penta FET X-3 Si (Li)) at 15 kV, 0.4 nA at a working distance of 10 mm. The fine grindings of the reduced samples were conducted on a sample pulverizer (4WR-3600, Kaifeng Yuda Machinery Equipment Co., Ltd. (Kaifeng, China)) with a power of 5 kw and a speed of 1500 r min^−1^. In addition, the chemical components of the raw material, anthracite, and the reduced ferronickel particles were analyzed by an inductively coupled plasma mass spectrometer (ICP-MS, 7700X, Agilent Co., Santa Clara, CA, USA). All of the tests were conducted three times to obtain the average values.

To further investigate the influence of various additives on the carbothermal reduction of the ferronickel concentrates, a thermodynamic analysis was carried out using FactSage version 6.4. The phase equilibrium was calculated by “Equilib function”, with the databases “FactPS”, “FToxid”, and “FShall”. The compound species of final products contained gas, liquid, and solid at a pressure of 1.0 atm. By inputting the initial compositions based on the sample’s composition, and setting the reaction temperature at 1250 °C, the phase equilibrium was calculated by thermodynamic calculations.

### 2.2. Raw Materials

#### 2.2.1. Laterite Nickel Ore

The laterite nickel ore used in this work was bought from Sulawesi, Indonesia. As displayed in Figure 1a, the raw laterite nickel ore exhibits a texture resembling clay, with lumps and a considerable amount of moisture, making it lack representativeness for sampling the laterite nickel ore directly. To address this, the raw laterite nickel ore was dried at 110 °C for 12 h, resulting in a mass loss of 27.9%. The dried sample was then subjected to a crushing process, followed by a thorough mixing of the crushed specimen. Subsequently, a fraction of the sample was finely ground in a vibration mill and then passed through a 0.125 mm sieve. The image of the fine ground laterite powder is shown in Figure 1b.

To investigate the chemical composition of laterite nickel ore, the finely ground laterite powder was dried at 105 °C until reaching a constant weight. Subsequently, comprehensive chemical analysis and XRD characterization were conducted. The XRD pattern shown in Figure 1c indicated that the laterite nickel ore predominantly consists of quartz (ICDD card No. 46-1045), lizardite (ICDD card No. 73-1336), magnetite (ICDD card No. 79-0417), and pargasite (ICDD card No. 89-7541). In addition, the chemical component of laterite powder was determined by ICP-MS, and the measured average element content is listed in Table 1, revealing a low nickel content of only 1.32 wt% and a total iron content of 15.87 wt%, alongside a remarkable abundance of magnesium oxide (15.10 wt%) and silicon dioxide (40.46 wt%).

#### 2.2.2. Reductants and Additives

Anthracite was employed as a reductant in this work, and its chemical composition determined by ICP-MS is shown in Table 2. Furthermore, calcium hydroxide (Ca(OH)_2_), calcium fluoride (CaF_2_), sodium sulfate (Na_2_SO_4_), and boric acid (H_3_BO_3_) were employed to lower the melting temperature and decrease its surface tension. Among them, the anthracite and slaked lime were bought from Wuhan Jiyesheng Chemical Co., Ltd. (Wuhan, China), and the other chemicals and reagents of at least analytical grade were purchased from Sinopharm Chemical Reagent Co., Ltd. (Shanghai, China).

#### 2.2.3. Thermodynamic Feasibility Analysis of Carbothermal Reduction

With anthracite as a reducing agent, the reduction of laterite nickel ore mainly consists of the fixed carbon reacting with CO_2_ to generate CO and the reductions of NiO, FeO, Fe_2_O_3_, and Fe_3_O_4_ with CO under normal pressure [24], which can be described as Equations (1)–(6):(1)$$\begin{array}{l} {\mathrm{CO}}_{2}+C=2\mathrm{CO}\\ \Delta G_{1}^{\theta }=166550-171T\;(J/\mathrm{mol}) \end{array}$$
(2)$$\begin{array}{l} {\mathrm{NiO}}_{\left(S\right)}{+C=\mathrm{Ni}}_{\left(S\right)}+\mathrm{CO}\\ \Delta G_{2}^{\theta }=118050-169.36T\;(J/\mathrm{mol}) \end{array}$$
(3)$$\begin{array}{l} {\mathrm{NiO}}_{\left(S\right)}{+\mathrm{CO}=\mathrm{Ni}}_{\left(S\right)}{+\mathrm{CO}}_{2}\\ \Delta G_{3}^{\theta }=-48500+1.64T\;(J/\mathrm{mol}) \end{array}$$
(4)$$\begin{array}{l} {3\mathrm{Fe}}_{2}O_{3(S)}{+\mathrm{CO}=2\mathrm{Fe}}_{3}O_{4(S)}{+\mathrm{CO}}_{2}\\ \Delta G_{4}^{\theta }=-52131-41T\;(J/\mathrm{mol}) \end{array}$$
(5)$$\begin{array}{l} {\mathrm{Fe}}_{3}O_{4(S)}{+\mathrm{CO}=3\mathrm{FeO}}_{\left(S\right)}{+\mathrm{CO}}_{2}\\ \Delta G_{5}^{\theta }=35380-40.16T\;(J/\mathrm{mol}) \end{array}$$
(6)$$\begin{array}{l} {\mathrm{FeO}}_{\left(S\right)}{+\mathrm{CO}=\mathrm{Fe}}_{\left(S\right)}{+\mathrm{CO}}_{2}\\ \Delta G_{6}^{\theta }=-22800+24.26T\;(J/\mathrm{mol}) \end{array}$$

The thermodynamic equilibrium curve of carbon, nickel oxide, and iron oxide based on reactions is shown in Figure 2, which can be divided into four temperature regions: the Fe_3_O_4_ and NiO stable region (t < 400 °C), the Fe_3_O_4_ and Ni stable region (400 °C < t < 656 °C), the FeO and Ni stable region (656 °C < t < 710 °C), and the Fe and Ni stable region (t > 710 °C). Therefore, the main factors affecting the reduction of iron and nickel oxides are reduction temperature, carbon content, and reduction atmosphere, and nickel and iron can theoretically exist in a metallic state when the reaction temperature is higher than 710 °C. However, this situation is hard to achieve in practical experimental conditions due to the low kinetic rate of the reaction of C with CO to form CO_2_ at 710 °C, and the reduction of iron and nickel oxides mainly depends on their solid phase reaction with carbon, making it difficult to realize the full reduction of nickel and iron. In addition, the low reaction temperature may also result in high energy consumption since the reaction time is prolonged. Therefore, increasing the temperature can significantly increase the CO formation and the iron and nickel reduction rates, thereby shortening the reaction time. The high temperature also facilitates the melting of the reduced ferronickel alloy and accelerates the separation of the ferronickel alloy from the ore. Based on our previous work and previously reported research [25,26,27,28,29], the reduction temperature in this work was set as 1250 °C.

### 2.3. Experimental Methods

#### 2.3.1. Sample Preparation

The laterite nickel ore samples were prepared by using 100 g laterite nickel ore powders, anthracite, and CaF_2_ as raw materials. The amount of added anthracite was based on the carbon ratio (n_C_/n_O_), which was defined as the ratio of oxygen atoms of metal oxides in the laterite nickel ore to carbon atoms in anthracite. Among them, anthracite was added with a carbon ratio of 1.2 or 1.4, and CaF_2_ was added in varying amounts of 6 wt%, 9 wt%, or 12 wt% of the laterite nickel ore powders, which are denoted as C1.2-F6, C1.2-F12, C1.4-F6, C1.4-F9, and C1.4-F12, respectively. The effect of slake lime on the carbothermal reduction of laterite nickel ore was investigated by adding the former to samples C1.2 and C1.2-F6 with a CaO/SiO_2_ ratio of 1.0, and the two samples are respectively denoted as C1.2-SL and C1.2-F6-SL. Two other additives, Na_2_SO_4_ and H_3_BO_3_, were added to C1.4-F6 or C1.4-F12 with additive amounts of 1 wt% or 2 wt% of the laterite nickel ore powders, and the samples are denoted as C1.4-F6-S1, C1.4-F12-B1, and C1.4-F12-B2, respectively. In addition, control samples were prepared by using laterite nickel ore powders and anthracite as starting materials with carbon ratios of 1.2 and 1.4, and they are respectively denoted as C1.2 and C1.4. The detailed raw material compositions of each sample are exhibited in Table 3.

#### 2.3.2. Carbothermal Reduction and Magnetic Separation

The carbothermal reduction experiment was conducted based on our previous research [25]. Typically, the laterite nickel ore, anthracite, and CaF_2_ were firstly mixed homogeneously, followed by the addition of water in a quantity of 10 wt% of the mixture. Subsequently, the mixtures were compressed into cylindrical specimens (φ 20 × h 30 mm, as depicted in Figure 3a). Then, the samples underwent drying in an oven at 110 °C overnight before being placed in a graphite crucible and heated to 1250 °C and soaked for 15 min under a N_2_ atmosphere in a 25 kW high-temperature carbon tube furnace (Figure 3b) with a heating rate of 120 °C/min. After the carbothermal reduction experiment, the samples were cooled to room temperature in the N_2_ atmosphere. Then, the samples were finely ground and underwent a magnetic separation before a chemical analysis to examine the iron and nickel content. In addition, the metal recoveries of Ni and Fe were calculated by Equations (7) and (8), respectively.
(7)$$\mathrm{Ni}\;\mathrm{recovery}=\frac{M_{\mathrm{Ni}}^{1}}{M_{\mathrm{Ni}}^{0}}\times \;100\%$$
(8)$$\mathrm{Fe}\;\mathrm{recovery}=\frac{M_{\mathrm{Fe}}^{1}}{M_{\mathrm{Fe}}^{0}}\times \;100\%$$
where $M_{\mathrm{Ni}}^{1}$ and $M_{\mathrm{Fe}}^{1}$ represent the qualities of Ni and Fe in reduced ferronickel concentrates after carbothermal reduction; $M_{\mathrm{Ni}}^{0}$ and $M_{\mathrm{Fe}}^{0}$ represent the qualities of Ni and Fe in lateritic nickel ore with carbon compacts before reduction.

## 3. Results and Discussion

### 3.1. The Morphology of As-Prepared Ferronickel Concentrates

As shown in Figure 4a–c, after the carbothermal reduction process, the melting degrees of the samples intensified as the CaF_2_ additive amount increased. The samples with CaF_2_ addition cannot hold their original cylindrical structures (Figure 4b,c), and their strengths were higher than those without CaF_2_. The microstructures of the samples C1.4, C1.4-F6, and C1.4-F12 are shown in Figure 4d–h, and it is obvious that the melting of samples C1.4-F6 and C1.4-F12 (Figure 4e–h) is comparable to that of sample C1.4 (Figure 4d). Moreover, a lot of metal particles with size of 9–25 μm were formed in sample C1.4-F12 (Figure 4g), which was rare in sample C1.4, indicating that the addition of CaF_2_ promoted the formation of molten slag, thus accelerated the agglomeration of metal particles. In addition, the EDS elemental mappings of sample C1.4-F12 shown in Figure 4i revealed the coexistence of Fe and Ni elements in the formed particles after the carbothermal reduction process, further demonstrating the formation of ferronickel particles, which is consistent with the previous works [14,17,25]. The presence of Mg, Si, and O elements within the same area suggested the formation of magnesium silicate. Furthermore, Cr_2_O_3_ and CaO may also exist in the reduced sample. The SEM results corroborated the formation of the ferronickel particles after the carbothermal reduction process, and the addition of CaF_2_ was advantageous to the metal polymerization and ferronickel particle agglomeration.

### 3.2. Iron and Nickel Contents of As-Prepared Ferronickel Concentrates after Fine Grinding and Magnetic Separation

Fine grinding of the as-prepared ferronickel concentrates is beneficial to improve the iron and nickel grades by separating the ferronickel particles from the concentrates. After carbothermal reduction, the as-prepared ferronickel concentrates were fine ground, and their particle size distribution is shown in Table 4 and Figure 5a. Obviously, the particle size of the sample decreased with the increase in grinding time, and the D_90_ of the sample was reduced to 26 μm after 240 s of fine grinding. Considering that the appropriate particle size facilitates the subsequent magnetic separation process, the optimal grinding time was selected as 240 s.

The fine ground samples were then subjected to a magnetic separation process to extract ferronickel concentrates for chemical analysis. Their iron and nickel contents of the samples are shown in Figure 5b. It is obvious that the contents of iron and nickel in sample C1.2-F6 exhibited significant improvement compared to the control sample C1.2 (iron: 43.26 wt% vs. 27.86 wt% and nickel: 5.63 wt% vs. 2.8 wt%), indicating that the introduction of CaF_2_ had a substantial enhancing effect on the formation of the ferronickel concentrate. However, samples C1.2-F6 and C1.4-F6 showed close iron and nickel contents, suggesting that the carbon ratio had little influence on the carbothermal reduction process. Furthermore, as the additive amount of CaF_2_ increased from 5.18 wt% to 7.57 wt% and then to 9.85 wt%, the initial grades of iron and nickel exhibited a decrease followed by an increase. Specifically, the iron grade decreased from 47.72 wt% to 40.11 wt% and then increased to 63.69 wt%, while the nickel grade decreased from 6.12 wt% to 5.51 wt% and then increased to 8.93 wt%. The improvement in the recovery of nickel and iron and concentrate grades may be attributed to the reduced surface tension of the mineral phase interface, which facilitated the growth of the iron and nickel solid solution [21]. The decrease in iron and nickel grades for the sample with 9 wt% CaF_2_ may have been caused by the entry of fine nonmagnetic material in the magnetic substance when the aggregation degree of the ferronickel was insufficient [25]. In conclusion, the ferronickel concentrates with the highest nickel grade of 8.93 wt% and iron grade of 63.96 wt% were obtained with the carbon ratio of 1.4 and the CaF_2_ additive amount of 9.85 wt%.

### 3.3. Effects of the Additive Type on the Iron and Nickel Contents of As-Prepared Ferronickel Concentrates

Additives were used to reduce the melting temperature and viscosity of the slag, thus enhancing the metallization, migration, and polymerization of iron and nickel elements. Moreover, slaked lime is often used to activate steel slag in the metallurgy industry [16]. In this work, after carbothermal reduction, samples C1.2-SL and C1.2-F6-SL, which possessed original cylindrical shapes, exhibited a low melting degree and lots of cracks on the surface (as displayed in Figure 6a,b), indicating that the addition of slaked lime reduced the cohesiveness of these samples, impeded the formation of molten slag, and consequently hindered the reduction of iron and nickel.

Figure 4e–h demonstrated that the addition of CaF_2_ facilitated the formation of molten slag, and then accelerated the reduction of iron and nickel from laterite nickel ore. Considering the relatively high cost of CaF_2_, Na_2_SO_4_ and H_3_BO_3_ were added to try to reduce the usage of CaF_2_. Regrettably, as shown in Figure 6c,d, both the quantity and particle size of ferronickel particles in sample C1.4-F6-S1 were obviously less than that of the sample C1.4-F6 in Figure 4e,f. The reason may be that the addition of Na_2_SO_4_ had an adverse impact on the melting degree of the samples and subsequently hindered the formation of molten slag and the reduction of laterite nickel ore as well.

When H_3_BO_3_ was added into the ore, after carbothermal reduction, substantial amounts of ferronickel particles with particle sizes of 1–10 μm were observed in Figure 6e,f, revealing that the addition of the additive significantly facilitated the formation of molten slag. Chemical analysis was employed to determine the iron and nickel contents of samples C1.4-F12-B1 and C1.4-F12-B2 and the results are illustrated in Figure 6g, demonstrating that the addition of H_3_BO_3_ led to a decrease in the iron and nickel grades compared to C1.4-F12 (iron: 46.09 wt% vs. 63.69 wt% and nickel: 6.71 wt% vs. 8.93 wt%). The decreased iron and nickel grades of C1.4-F12-B2 may be attributed to the difficult separation of the small ferronickel particles since the size is far below the D_90_ of sample C1.4-F12 after 240 s of fine grinding (<10 μm vs. 26 μm). Thus, the subsequent separation of ferronickel particles from the molten slag during the magnetic separation process became challenging [28].

To our knowledge, the carbothermal reduction of laterite nickel ore with the presence or carbon and additives is widely reported. However, the nickel and iron grades, as well as their recoveries in the present work, were significantly higher than these in most recently reported works, which are compared in Table 5, indicating the distinguished nickel reduction performance of the present carbothermal reduction method. Moreover, it is worth emphasizing that the reduced ferronickel concentrates in this work can be easily separated using a natural magnet, while most reported works usually required an extra electromagnetic field for separation.

In a word, after carbothermal reduction and magnetic separation, high-grade magnetically separable ferronickel concentrates were successfully prepared by using laterite nickel ore, anthracite, and CaF_2_ as raw materials, and the sample C1.4-F12 exhibited the highest iron grade of 63.96 wt% and nickel grade of 8.93 wt%. The optimal processing conditions for preparation of the present high-grade ferronickel concentrates were as follows: the carbon ratio of 1.4, CaF_2_ addition of 9.85 wt%, and fine grinding time of 240 s. Moreover, the present carbothermal reduction process for laterite nickel ores can be extended to practical large-scale production easily.

### 3.4. Thermodynamic Equilibrium and Melting Temperature Calculation

To further investigate the influence of various additives on the carbothermal reduction of the ferronickel concentrates, a thermodynamic analysis was carried out using FactSage version 6.4 following the methods mentioned in Section 2.2.3. The results are presented in Table 6, indicating that the reductions of the iron oxide and nickel oxide are thermally accessible in the current experimental conditions.

Since the iron oxide and nickel oxide reductions of these samples were all thermodynamically feasible, the main factor affecting the reduction rate of iron and nickel should be the differences in the kinetic reaction rate, which was influenced by the additives. Therefore, the equilibrium and phase transition temperature (also referred to as melting temperature) of each sample within the range of 400–1500 °C were calculated. As shown in Figure 7, the addition of CaF_2_, Na_2_SO_4_, and H_3_BO_3_ all resulted in a decrease in the melting temperature. However, the case with slaked lime only had a slight decrease compared to other additives, and the melting temperatures of samples C1.2-SL and C1.2-F6-SL were both higher than the experimental temperature of 1250 °C. The most significant decrease in melting temperature (235 °C) was obtained for the case with CaF_2_ of 9.85 wt%, demonstrating the conducive role of CaF_2_ in the formation of the ferronickel concentrate [29]. Moreover, sample C1.4-F12-B2 exhibited the lowest melting temperature of 950 °C, indicating that the addition of H_3_BO_3_ contributed to the formation of molten slag, which was aligned with the experimental results in Figure 7e,f.

In summary, the thermodynamic calculation indicated that the addition of CaF_2_ exerted a predominant influence on the reduction of iron and nickel by facilitating the formation of molten slag, which was beneficial to the reduction and formation of ferronickel concentrate in kinetics. The addition of Na_2_SO_4_ and H_3_BO_3_ can also promote molten slag formation while the slaked lime only had a limited effect.


## 4. Conclusions

In this work, the influence of different additives on the carbothermal reduction of laterite nickel ore and the effect of fine grinding time on the iron and nickel contents in the ferronickel concentrate were investigated. Based on experimental results, the conclusions can be drawn as follows:The addition of Ca(OH)_2_ and Na_2_SO_4_ played negative roles in the formation of iron and nickel, while CaF_2_ and H_3_BO_3_ were beneficial to that. The higher the amount of CaF_2_ and H_3_BO_3_ added, the higher the grade of the obtained ferronickel concentrate. Thermodynamic calculation also confirmed that the addition of CaF_2_ can reduce the melting temperature of the sample, which facilitated the reduction of iron and nickel.The thermodynamic calculation revealed that the simultaneous addition of CaF_2_ and H_3_BO_3_ can significantly reduce the formation temperature of the melting phase in the carbothermal reduction process, facilitating the reduction of iron and nickel, which was aligned with experimental results. However, the sizes of reduced ferronickel alloy particles were too small (<10 μm), making it difficult to separate them from the ore even after fine grinding, thus decreasing the recovery of the ferronickel alloy. Conversely, in the cases of only adding CaF_2_, natural magnetism was sufficient to separate the ferronickel particles from the product after fine grinding, eliminating the need for additional electromagnetic fields and thereby reducing energy consumption.Increasing the fine grinding time was beneficial for improving the grade of the ferronickel concentrate. The optimal fine grinding time of 240 s was conductive to the magnetic separation of ferronickel particles.High-grade ferronickel concentrates with nickel content of 8.93 wt% and iron content of 63.96 wt% were obtained after reducing laterite nickel ore at 1250 °C for 15 min with a carbon ratio of 1.4 and 9.85 wt% CaF_2_, which can serve as raw materials in the blast furnace iron smelting, stainless steel, alloy manufacturing, and electroplating industries.The metal recoveries of nickel and iron in this work reached 96.28% and 82.14%, respectively, which offers guidance on effectively exploiting and utilizing low-grade laterite nickel ore to maintain the nickel supply. In addition, the design and implementation of a larger-scale carbothermal reduction process for laterite nickel ores are expected to be achieved.


## Figures and Tables

**Figure 1 materials-16-07132-f001:**
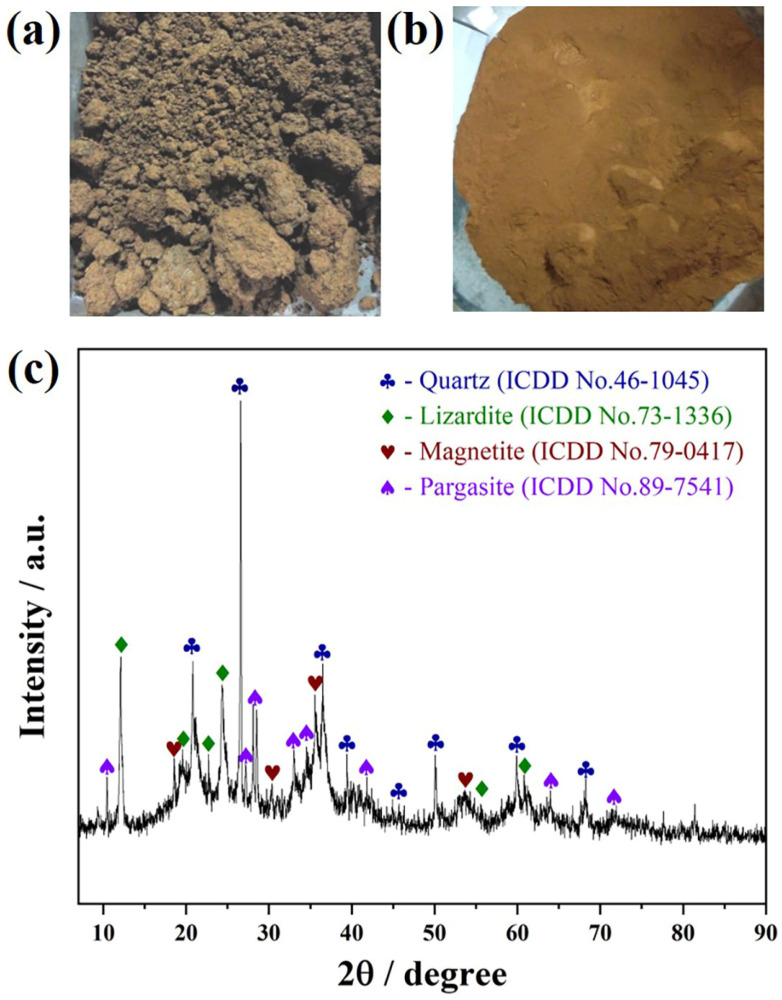
Photograph of (**a**) raw laterite nickel ore and (**b**) laterite nickel ore fine powders, and (**c**) XRD pattern of laterite nickel ore fine powders.

**Figure 2 materials-16-07132-f002:**
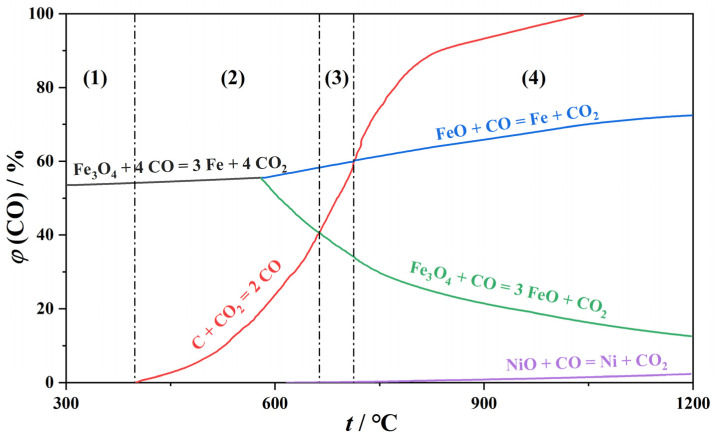
Thermodynamic equilibrium diagram of iron oxide and nickel oxide reduced by carbon. ((1): Fe_3_O_4_ and NiO stable region (t < 400 °C), (2): Fe_3_O_4_ and Ni stable region (400 °C < t < 656 °C), (3): FeO and Ni stable region (656 °C < t < 710 °C), and (4): Fe and Ni stable region (t > 710 °C))

**Figure 3 materials-16-07132-f003:**
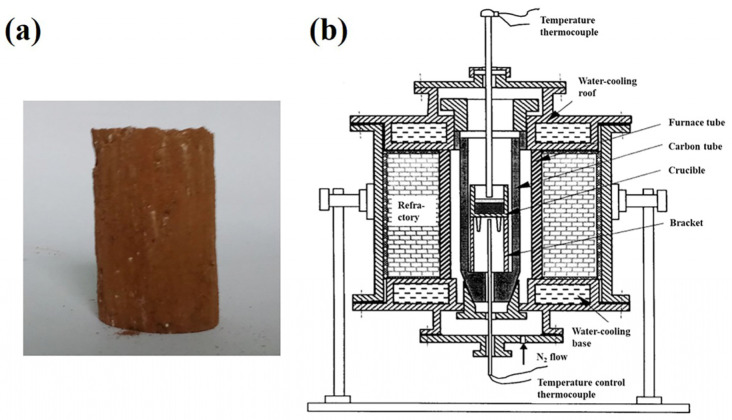
(**a**) Photograph of the pressed cylindrical sample and (**b**) schematic diagram of high-temperature carbon tube furnace.

**Figure 4 materials-16-07132-f004:**
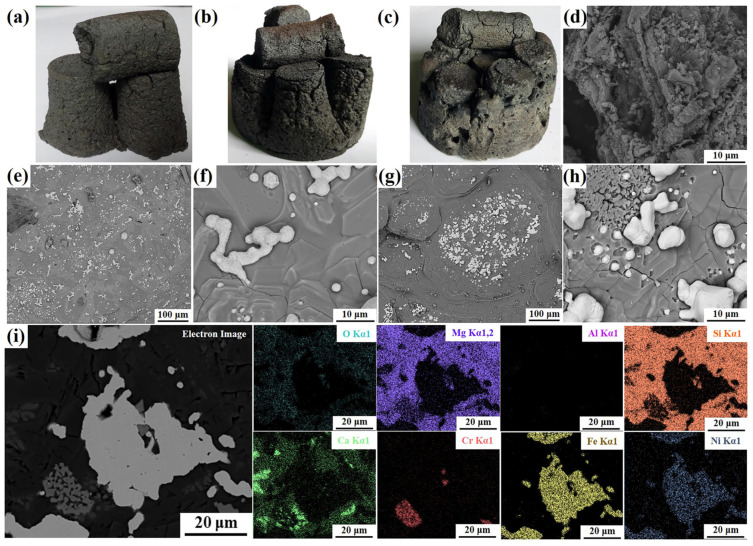
Photographs of (**a**) sample C1.4, (**b**) sample C1.4-F6, and (**c**) sample C1.4-F12; SEM images of (**d**) sample C1.4, (**e**,**f**) sample C1.4-F6, and (**g**,**h**) sample C1.4-F12; and (**i**) EDS elemental mappings of sample C1.4-F12.

**Figure 5 materials-16-07132-f005:**
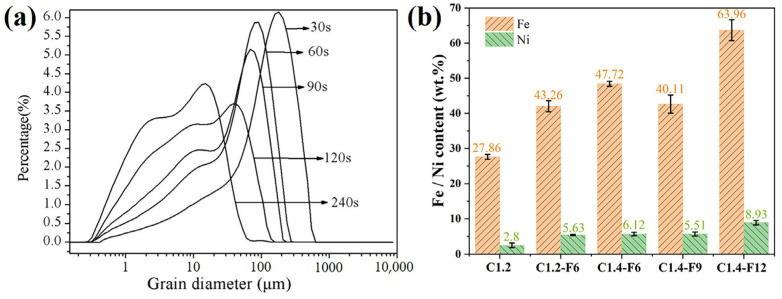
(**a**) Particle size distributions of as-prepared ferronickel concentrates with different fine grinding time, and (**b**) chemical analysis of ferronickel concentrates with different CaF_2_ additive amounts.

**Figure 6 materials-16-07132-f006:**
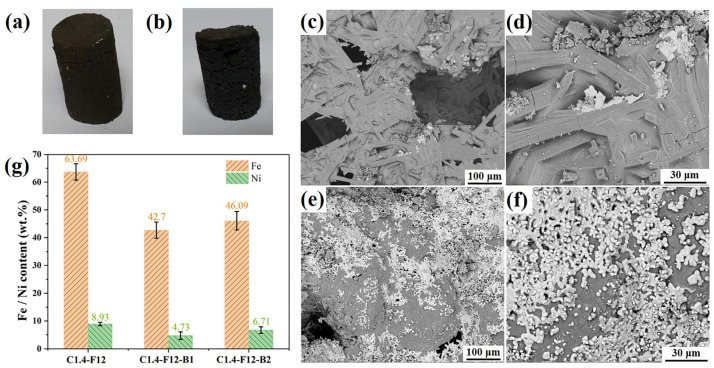
Photographs of samples (**a**) C1.2-SL and (**b**) C1.2-F6-SL after carbothermal reduction; and SEM images of (**c**,**d**) sample C1.4-F6-S1, (**e**,**f**) sample C1.4-F12-B1; and (**g**) chemical analysis of samples C1.4-F12, C1.4-F12-B1, and C1.4-F12-B2.

**Figure 7 materials-16-07132-f007:**
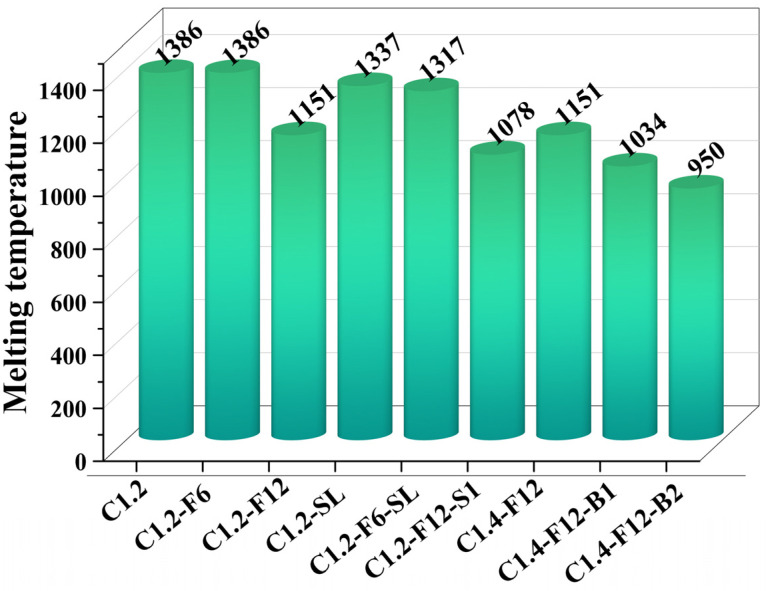
The calculated melting temperature of the samples with different types and amounts of additives.

**Table 1 materials-16-07132-t001:** Chemical analysis of laterite nickel ore powders.

Component	Fe_total_	Ni_total_	CaO	MgO	SiO_2_	Al_2_O_3_	P_2_O_5_	Na_2_O	K_2_O	TiO_2_	S	LOI ^1^
**Content (wt%)**	15.87	1.32	0.80	15.10	40.46	2.79	0.02	0.23	0.29	0.02	0.02	15.24

^1^ LOI: Loss on ignition.

**Table 2 materials-16-07132-t002:** Chemical analysis of anthracite and slaked lime.

	Component (wt%)	CaO	MgO	SiO_2_	Al_2_O_3_	C	P_2_O_5_	S	Na_2_O	K_2_O	TiO_2_
Sample											
Anthracite		1.28	0.35	6.09	2.68	76.56	0.23	0.54	0.02	0.41	0.15
Slaked lime		67.52	4.56	0.56	-	2.11	0.01	-	0.04	0.06	-

**Table 3 materials-16-07132-t003:** Raw material compositions of laterite nickel ore samples.

	Component (wt%)	Laterite	Carbon Ratio	Anthracite	Slaked Lime	CaF_2_	Na_2_SO_4_	H_3_BO_3_
Sample								
C1.2		92.21	1.2	7.79	-	-	-	-
C1.4		91.03	1.4	8.97	-	-	-	-
C1.2-F6		87.37	1.2	7.38	-	5.24	-	-
C1.2-F12		83.02	1.2	7.02	-	9.96	-	-
C1.4-F6		86.32	1.4	8.50	-	5.18	-	-
C1.4-F9		84.14	1.4	8.29	-	7.57	-	-
C1.4-F12		82.07	1.4	8.08	-	9.85	-	-
C1.2-SL		50.84	1.2	4.30	44.86	-	-	-
C1.2-F6-SL		49.34	1.2	4.17	43.53	2.96	-	-
C1.4-F6-S1		85.58	1.4	8.43	-	5.13	0.86	-
C1.4-F12-B1		81.40	1.4	8.02	-	9.77	-	0.81
C1.4-F12-B2		80.74	1.4	7.95	-	9.69	-	1.61

**Table 4 materials-16-07132-t004:** Particle size distribution of as-prepared ferronickel concentrates after fine grinding.

	Grinding Time (s)	30	60	90	120	240
Distribution (μm)						
D_10_		10	4	3	2	1
D_50_		52	52	32	12	6
D_90_		339	147	114	63	26

**Table 5 materials-16-07132-t005:** Comparison of the nickel and iron reduction performance with some reported works via the carbothermal reduction method.

Raw Material	Reduction Temperature (°C)	Additive	Metal Grade in Raw Material (wt%)		Metal Content in Reduction Product (wt%)		Metal Recovery (%)		Ref.
			Ni	Fe	Ni	Fe	Ni	Fe	
Saprolitic laterite ore	1300	CaF_2_	0.8	10.9	6.1	68.2	90	75.7	[24]
	1200	CaSO_4_	1.4	16.18	8.08	79.98	92.6	79.9	[22]
Laterite nickel ore	1250	NaCl	1.13	35.79	8.15	64.28	97.76	-	[23]
	1500	CaF_2_	1.27	46.54	1.79	89.65	71.3	59.96	[25]
	1550	-	1.29	16.31	10.02	84.02	95.51	-	[5]
	1275	-	2.26	14.24	6.96	34.74	94.06	80.44	[6]
	1200	CaF_2_	1.4	16.18	7.1	68.5	84.14	70.24	[1]
	1250	CaF_2_	1.06	11.56	8.39	67.7	98.54	71.73	[4]
	1250	CaF_2_	1.32	15.87	8.93	63.96	96.28	82.14	This work

**Table 6 materials-16-07132-t006:** Iron and nickel contents calculated by thermodynamic analysis and the related reduction ratios in the phase equilibrium condition at 1250 °C.

Sample	Fe Content (g)	Reduction Ratio of Fe * (%)	Ni Content (g)	Reduction Ratio of Ni * (%)
C1.2-F6	13.26	99.98	1.38	100
C1.2-F12	13.26	99.98	1.38	100
C1.4-F6	13.26	99.98	1.38	100
C1.4-F9	13.26	99.98	1.38	100
C1.4-F12	13.26	99.98	1.38	100
C1.2-F12-S1	13.25	99.90	1.38	100
C1.4-F12-B1	13.26	100	1.38	100
C1.4-F12-B2	13.26	99.98	1.38	100

$*\;\mathrm{Reduction}\;\mathrm{ratio}\;\mathrm{of}\;\mathrm{Fe}/\mathrm{Ni}=\frac{\mathrm{Thermodynamic}\;\mathrm{calculated}\;\mathrm{Fe}/\mathrm{Ni}\;\mathrm{content}}{\mathrm{Fe}/\mathrm{Ni}\;\mathrm{content}\;\mathrm{in}\;\mathrm{raw}\;\mathrm{materials}}$.

## Data Availability

Data will be made available on request.

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
