# Peer review of "High-Grade Ferronickel Concentrates Prepared from Laterite Nickel Ore by a Carbothermal Reduction and Magnetic Separation Method"

_materials, 2023, doi:10.3390/ma16227132_

Round 1
Reviewer 1 Report
Comments and Suggestions for Authors
Dear Authors:
Congratulations on the important experimental work you have done in your research.
I submit the following comments intended to contribute to your manuscript:
- It seems incongruous that you highlight the need to achieve carbon neutrality but use carbon precisely as a reducer. Is there any justification for this? Couldn't a reduction process be designed based on sustainability based on the use of another reducing agent?
Its material has a grade of 1.32% Ni and an iron grade of 15.87%. Why do they present it as a low-grade material?
Table 1 shows the chemical analysis of nickel ore. What technique was used? How reliable and reproducible are these analyses? Do you know the remaining unidentified percentage that is considerably high?
Line 114: Please consider changing were exhibited to are shown or are presented.
Line 121: What the symbol represents in the dimensions.
Line 142: In terms of what are samples C1.4-F6 more clearly compared to sample C1.4?
Was the magnetic concentration operation evaluated with the original material without reduction?
In Table 3, no experiment contains an addition of 12% CaF2, as stated in line 196.
Why would polymerization of iron and nickel occur? Is it common for metals to polymerize?
Wouldn't it seem expected that the most intense conditions for the operation were precisely those that gave them the highest percentage reduction?
Is the reduced material composed of 8.93% Ni and 63.96% Fe pure enough to be commercialized and used in the applications for which it is intended?
What other aspects should be considered to advance the design and implementation of a larger-scale process?
Best regards.
Comments on the Quality of English LanguageDear Authors:
I suggest reviewing your manuscript's writing style beyond the possibility that it presents serious problems in grammar or writing.
Reviewer 2 Report
Comments and Suggestions for Authors
The proposed study addresses the production of high-grade ferronickel concentrates by a carbothermal reduction and magnetic separation method from low-grade laterite nickel ore. The manuscript is concise and clear, and the results are explained, however, certain corrections are needed, which I list below:
1. In part 2, Materials and Methods, add a segment where all the methods applied in this study will be summarized, describe the methods of chemical analysis and according to what standards it was made, and list all the used devices. The contents of Na2O, K2O, and TiO2 are also missing in the chemical analysis.
2. If the method from part 2.2.2 was used for the first time, highlight it, if not refer to the relevant literature.
3. Are standard deviations available for the results?
4. Compare results with relevant literature or similar procedures.
5. Emphasize the practical contribution of this study, and give a hint of concrete application in industrial conditions.
Reviewer 3 Report
Comments and Suggestions for Authors# Introduction:
- The novelty is not clear. The reduction of nickeliferous laterites, in the presence or carbon or hydrogen, is a widely reported subject. The inclusion of additives is also well documented in the literature. The authors must support their novelty based on the reaction system combination, their potential regarding selective recovery of strategic metals and a clear overview of how such information is new to the extractive metallurgy community.
- Considering such aspect, the literature review made by the authors is minimal. Only 14 citations for a raw material of worldwide interest and with many options of metal extraction explored recently is below average. The authors must significantly improve this characteristic of their manuscript.
# Materials Characterization:
- All equipment used in this part must be detailed in terms of equipment manufacturer, model and methodology. Moreover, in Fig1, it is impossible to assess the XRD quality without any counts number. In addition, there are many peaks that are not related to any mineralogical phase. The relationship between signal and detected peaks is not presented and the authors does not cover a quantitative assessment of the XRD spectrum. The XRF, in Tab1, must be presented in terms of elements and not oxides. A correction must be made for the reader to have a proper understanding of the mineral sample characteristics. The same issue appears in Tab2 for the reagents of the reaction system.
- The authors did mention a SEM equipment for the characterization. However, such information must not be included in the experimental procedure related to the chemical processing of the laterite. It is imperative that the authors present their methodology in a orthodox approach, separating what is related to materials characterization from used materials/chemicals and their experimental extraction procedure.
# Chemical Reagents:
- All chemicals must be detailed in terms of manufacturer and grade, in case of synthetic solids, or mineral origin, in case of natural occurrences.
- The issue in Tab2 reported previously must be solved.
# Experimental Procedure:
- The authors must explain the option regarding the used proportions between laterite and chemical reagents.
- The authors must explain why they opted by the 1250 °C as reaction temperature for their study. If not based on thermodynamics simulations or literature review, such information has no scientific meaning. It is interesting to assess some Gibbs Free Energy information as well as Equilibrium Composition to define the reaction system condition.
# Results:
- It is not clear the procedure for quantification of elements in the reaction products. Typical characterization techniques are missing.
- The methodology for thermodynamics information regarding the phase equilibrium is not reported in the Materials and Methods section.
# Overall:
- The manuscript presents an interesting reaction system which could not be assessed as novel information based on the little literature review.
- The methodology presentation is incomplete and not organized.
- The document must go through another round of writing and editing prior to a new submission in the future. In the present format it cannot be published in a prestigious journal as Materials-MDPI.
- The authors should consider including more results also to support their observations. The amount of information reported is not sufficient in the current state for publication.
Reviewer 4 Report
Comments and Suggestions for Authors
Manuscript ID: materials-2678211
Type of manuscript: Article
Title: High-grade ferronickel concentrates prepared from laterite nickel ore by a carbothermal reduction and magnetic separation method
Authors: Jingzhe Zhang, Chang Cao, Zhengliang Xue, Faliang Li, Shaoping Li, Hongjuan Duan *, Haijun Zhang *
This paper describes new separation method of ferronickel. This issue is very important in current industry.
However, I found many questionable points, therefore, it should be revised as major revision to the next step.
Table 1
It does not say how these values were obtained from XRD pattern.
I think it is impossible to obtain these values with these data since all these intensities were quite weak and the signal-to-noise ratio is very poor. Also, there must be some errors in this value depending on the sample or where the sample was taken, so standard deviations at least three subsequent measurements are necessary.
Table 2
It does not say how this value was obtained. The only thing it says is "chemical analysis".
As in Table 1, the standard deviation of at least three measurements should be described.
Page 4 line 129
Please indicate the specific method "a chemical analysis to examine the iron and nickel content.
Results and discussion
I don't think it is good that you are discussing this manuscript without citing mostly previous papers. The citation references are only 18 it was too small numbers.
There are only [12] (page 7, line 191), [17] (page 8, line 226), and [18] (page 9, line 259) in this part.
Overall
I think this paper is too short. Does it meet the submission rules of this journal?
There is little description of the experimental method, and I am not sure if the reader will be able to follow this method. You should reconsider.
I think 18 citations is also too few. The authors should cite past papers and show how your results could differ from the past purification methods.
Comments on the Quality of English LanguageSome careless errors were found.
Reviewer 5 Report
Comments and Suggestions for Authors
This study focused on enhancing the extraction of nickel from low-grade laterite nickel ore, which mostly contains nickel oxide in complex compositions. The researchers used a carbothermal reduction and magnetic separation process with laterite nickel ore and anthracite. They experimented with different additives (CaF2, Na2SO4, and H3BO3), carbon ratio, and grinding time to optimize the recovery and grade of ferronickel concentrates. CaF2 was identified as the primary additive, positively influencing the aggregation and growth of ferronickel particles. Under optimal conditions (12 wt.% CaF2, carbon ratio of 1.4, and 240 s grinding time), they successfully obtained high-grade magnetically separable ferronickel concentrate with 8.93 wt.% nickel and 63.96 wt.% iron. The study provides a practical method for efficiently recovering and utilizing iron and nickel from low-grade laterite nickel ore, contributing to sustainable nickel resource extraction.
The manuscript is well written and shows interesting results and discussions. However, the novelty of the research is not clearly emphasized. Below is a list of suggestive revisions that might help improve the manuscript.
1) “Abstract” needs to explain what, why and how the research was implemented. At its current, it reads as summary of the results. I would suggest adding a few sentences where the novelty of the work and how it added to the prior art would be clearly emphasized.
2) “Introduction” needs to cover all the literature related to the topic completely. It reads short at its current status. Furthermore, in the very last paragraph, it needs to clearly include a problem statement followed by the clear emphasis on the novelty of the work and how it added to the prior art.
3) “Experimental section” needs include the ISO/ASTM standard procedure citation for metallography, mechanical testing, chemical analysis and so forth. Moreover, it needs to include detailed explanation of metallography (such as the grinding paper types, forces) as well as technicality of the electron microscopy/EDS (such as the working distance, kV, and so forth).
4) The discussion section, in general, is a little too short and may need to be strengthened using more in-depth discussion.
Round 2
Reviewer 1 Report
Comments and Suggestions for Authors
Dear Authors:
After reading the new version of your manuscript, I consider it to be strong enough to be considered for publication. It would have been interesting if your article was as strong as your answers. Please review the ambiguity and interchangeability of particle agglomeration with metal polymerization. The last reply about the potential of the process to be applied on a larger scale would have been interesting if you had included it in your manuscript in a summarized way to contribute to scientific and technological knowledge in the area.
Reviewer 3 Report
Comments and Suggestions for Authors
The authors improved their manuscript.
All equipment used in the materials characterization of this manuscript must be detailed by means of manufacturer and model. This must be presented prior to the raw-materials characterization. The suggestion here is that the authors present first their item of Materials Characterization and then present the item Raw-Material.
Figure 2 must have authorization for publication from the publishing company and maybe the authors. The suggestion is that the authors made their on version of that using a typical software for thermodynamics calculations sucha as: ThermoCalc, FactSage or HSC Chemistry. This is also a suggestion.
The document must be go minor revision only.
Reviewer 4 Report
Comments and Suggestions for Authors
Now, it will be sufficient for the publication in the current version.
Comments on the Quality of English LanguagePlease check over the manuscript in the proof check version.
Author Response
Dear Reviewer,
We would like to express our sincerest gratitude for your valuable comments on our work. Your attention and constructive suggestions have been immensely helpful to us in improving the quality of our work. We truly appreciate the time and effort that you have dedicated to our manuscript.
Thank you again for your attention and guidance.
Reviewer 5 Report
Comments and Suggestions for Authors
The previously provided comments were satisfactorily addressed.
Author Response

(The authors gave the same response as above.)
